# Encapsulation Effect on the In Vitro Bioaccessibility of Sacha Inchi Oil (*Plukenetia volubilis* L.) by Soft Capsules Composed of Gelatin and Cactus Mucilage Biopolymers

**DOI:** 10.3390/polym12091995

**Published:** 2020-09-02

**Authors:** María Carolina Otálora, Robinson Camelo, Andrea Wilches-Torres, Agobardo Cárdenas-Chaparro, Jovanny A. Gómez Castaño

**Affiliations:** 1Grupo de Investigación en Ciencias Básicas (NÚCLEO), Facultad de Ciencias e Ingeniería, Universidad de Boyacá, 150001 Tunja, Boyacá, Colombia; andreawilches@uniboyaca.edu.co; 2Grupo Química-Física Molecular y Modelamiento Computacional (QUIMOL), Facultad de Ciencias, Universidad Pedagógica y Tecnológica de Colombia (UPTC), 150001 Tunja, Boyacá, Colombia; luis.camelo@uptc.edu.co (R.C.); agobardo.cardenas01@uptc.edu.co (A.C.-C.)

**Keywords:** softgels, mucilage, biopolymers, in vitro digestion, bioaccessibility

## Abstract

Sacha inchi (*Plukenetia volubilis* L.) seed oil is a rich source of polyunsaturated fatty acids (PUFAs) that are beneficial for human health, whose nutritional efficacy is limited because of its low water solubility and labile bioaccessibility (compositional integrity). In this work, the encapsulation effect, using blended softgels of gelatin (G) and cactus mucilage (CM) biopolymers, on the PUFAs’ bioaccessibility of *P. volubilis* seed oil was evaluated during in vitro simulated digestive processes (mouth, gastric, and intestinal). Gas chromatography–mass spectrometry (GC–MS) and gas chromatography with a flame ionization detector (GC–FID) were used for determining the chemical composition of *P. volubilis* seed oil both before and after in vitro digestion. The most abundant compounds in the undigested samples were α-linolenic, linoleic, and oleic acids with 59.23, 33.46, and 0.57 (g/100 g), respectively. The bioaccessibility of α-linolenic, linoleic, and oleic acid was found to be 1.70%, 1.46%, and 35.8%, respectively, along with the presence of some oxidation products. G/CM soft capsules are capable of limiting the in vitro bioaccessibility of PUFAs because of the low mucilage ratio in their matrix, which influences the enzymatic hydrolysis of gelatin, thus increasing the release of the polyunsaturated content during the simulated digestion.

## 1. Introduction

Sacha inchi (*Plukenetia volubilis* L.) is a plant belonging to the Euphorbiaceae family that grows in the Amazon rainforest in Northeastern Peru and Northwestern Brazil. It has aroused interest in the food industry because of the high quantity and quality of the edible oils contained in its seeds. Its fruit consists of a star-shaped capsule that is approximately 3–5 cm in size, and normally contains between four and six dark brown edible oval seeds that are 1.5–2 cm in size [1]. The seeds of this plant are an excellent source of edible oil (41–54%). This oil contains lipids (35–60%), free fatty acids (1.2%), and phospholipids (0.8%) [2]. The high nutritional value of sacha inchi oil is due to its high polyunsaturated fatty acid (PUFA) and monounsaturated fatty acid (MUFA) content, which varies between 77.5% and 84.4%, and 8.4% and 13.2%, respectively [2,3,4]. α-Linolenic acid (ALA; C18:3, ω-3) is the major fatty acid (47–51%), followed by linoleic acid (LA; C18:2, ω-6, 34–37%) and oleic acid (ω-9, 9–10%) [2]. These fatty acids are considered beneficial because of their antioxidant, antithrombotic, antidyslipidemic, and anticancer effects [5,6,7]. Although PUFAs are beneficial, they are very sensitive to oxidative damage when exposed to oxygen, and are affected by heat, light, and humidity [8]. Therefore, it is essential to establish adequate systems for the transport and encapsulation of fatty acids, while maintaining their nutritional properties until they are released within the body.

Encapsulation with hydrocolloid biopolymers is an effective and widely used technique in the food industry in order to protect dietary supplements against oxidation and loss of nutritional value. Its effectiveness is as a result of the hermeticity of its walls and the safe supply of bioactive compounds, due to its rapid disintegration in biological fluids at body temperature [9,10]. This characteristic makes soft capsules (or softgels) one of the most widely used encapsulation methods because of their safe and nutritionally-accepted means of delivering aqueous liquid or semi-solid dosage formulations. Although there are several studies on the encapsulation of omega-3 fatty acids using different techniques [11,12,13,14], the use of soft capsules for the encapsulation of polyunsaturated fatty acids in scientific studies is scarce [15,16].

Softgels are generally manufactured with animal-derived gelatin (G), water, non-volatile plasticizers, and minor additives such as opacifiers and dyes [17,18]. The choice of G as a traditional material in the walls of softgels is related to its biodegradable nature and its ability to form thermo-responsive hydrogels [19,20]. An emerging trend in the food industry has recently sought to partially or completely replace G with other non-animal natural hydrocolloids. This has allowed plant mucilage to emerge as an attractive structural biopolymer alternative to soft capsules [21]. The mucilage extracted from the cladodes of *Opuntia ficus-indica* is a heteropolysaccharide matrix that has proven to be a suitable natural structuring material in soft capsules because of its high fiber content and desirable functional properties [22,23,24].

Bioaccessibility is defined as the amount of a bioactive compound that can cross the intestinal barrier as a result of its release from the matrix by the action of digestive enzymes [25]. Therefore, the stability of the encapsulated polyunsaturated fatty acids that are released into the gastrointestinal tract and are available for intestinal absorption is directly related to the biocompatibility and biodegradability rates of the capsule matrix [26]. As a result, the behavior of the capsule structure in relation to changes in pH and the presence of digestive enzymes and bile salts is an important factor that should be properly evaluated. This makes in vitro methods that simulate gastrointestinal media particularly advantageous for the quantification of the biocompatibility and biodegradability of soft capsules containing such nutrients. The advantages of in vitro methods include speed, cost efficiency, and a lack of ethical restrictions when compared to in vivo methods [27]. The use of in vitro conditions provides the opportunity to evaluate the suitability of a matrix’s ability to carry functional compounds of interest for the food and pharmaceutical industries. The bioaccessibility of polyunsaturated fatty acids has been evaluated using in vitro models in different studies [28,29].

The primary interest of this study was to determine the effect of gelatin/cactus mucilage as a softgel wall material in the in vitro digestion of encapsulated sacha inchi oil (SIO). The bioaccessibility analysis was performed using a comparative study that involved the quantification and identification of bioactive compounds using gas chromatography with a mass spectrometry (MS) detector and a flame ionization (FID) detector. The quantification was performed both before and after different laboratory-scale digestion processes. To the best of our knowledge, this study is the first evaluation of soft capsule wall matrices in the bioaccessibility of edible SIOs, which may be a topic of relevant interest to the food supplement industry.

## 2. Materials and Methods

### 2.1. Materials and Reagent

Sacha inchi fruits were supplied by local farmers in Miraflores (Boyacá, Colombia). The seeds were selected manually by discarding those that presented physical damage. They were packaged in polyethylene bags and stored at 18 °C until use. Cactus mucilage (CM) was extracted from the cladodes of *Opuntia ficus-indica* provided by local farmers in Duitama (Boyacá, Colombia), following the methodology reported by Quinzio et al. [30], and were used without further purification. Food grade gelatin (Type B, bloom strength 285, pI 4.2–6.5, Mw 40,000–50,000 Da, and ~99% purity) was provided by Gelco (Medellín, Antioquia, Colombia). Glycerol was provided by Merck (Darmstadt, Germany) for use as a plasticizer. Commercial samples of digestive enzymes and bile salts (α-amylase, pepsin from porcine gastric mucosa, and pancreatin from porcine pancreas) were obtained from Sigma–Aldrich (Auckland, New Zealand).

### 2.2. Sacha Inchi Oil (SIO) Extraction

The SIO extraction was performed using the Soxhlet methodology with chloroform as a solvent. Soxhlet extraction was selected as the most conventional, economical, and easiest process to implement. To maximize the SIO extraction, approximately 15 g of seed material was ground in an analytical mill (IKA A11 basic S1). Then, 1 g of seed sample was placed in a cellulose cartridge (33 × 80 mm) for extraction using a Soxhlet extractor SER 148/3 Velp Scientifica (Usmate Velate, Italy) for 1.5 h with 60 mL of solvent.

### 2.3. Characterization Sacha Inchi Oil

#### 2.3.1. Fatty Acid Identification with Gas Chromatography Coupled to Mass Spectrometry (GC–MS)

The chemical composition of the SIO sample was determined by gas chromatography coupled to mass spectrometry (GC/MS) using a 6890 N GC–MS instrument (Agilent Technologies Inc., Palo Alto, CA, USA) coupled to an Agilent 5973 N inert mass selective (IMS) detector. A capillary column DB-1MS (30 m × 0.25 mm ID with 0.25 μm film thickness) was employed for the analysis. The elution program started at a temperature of 70 °C, held for 2 min, and then increased to 320 °C at a speed of 8 °C/min and was held for 29 min. Both the IMS detector and the injector port temperatures were 320 and 250 °C, respectively. The injection volume used was 3.0 μL, and helium was used as the carrier gas at a flow rate of 1 mL/min in a splitless mode. The components were identified using a commercial library higher than 85% (WileyW9N08, Mass Spectral Database of the National Institute of Standards and Technology (NIST)).

#### 2.3.2. Fatty Acid Profile with Gas Chromatography with a Flame Ionization Detector (GC–FID)

The SIO sample was also analyzed using a 6890 N GC–FID instrument (Agilent Technologies Inc.). A capillary column DB-225 (60 m × 0.25 mm ID with 0.25 μm film thickness) was employed for the analysis. The elution program started at a temperature of 75 °C and increased to 220 °C at a speed of 5 °C/min for 50 min. Both the detector and injector port temperatures were 220 and 250 °C, respectively. The injection volume used was 0.2 μL, and helium was used as the carrier gas at a flow rate of 1 mL/min in a 100:1 split mode.

### 2.4. Soft Capsule Preparation with SIO Inclusion

The soft capsule design was developed according to the method reported by Camelo et al. [21]. CM (1.0 g) and food grade G (21.0 g) were separately dissolved in 100 mL of distilled water at 18 °C and 40 °C for 2 h and 30 min, respectively. Both solutions were constantly stirred at 300 rpm using a magnetic stirrer (C-MAG HS 7S000, IKA, Staufen im Breisgau, Germany) in order to ensure complete solubilization. Afterwards, the gelatin solution was mixed separately with glycerol (Gly) at a concentration of 15% (*w*/*v*) at 60 °C for 2 h to remove residual air bubbles and to obtain a homogeneous solution. The ratio of G/CM (3:1 *w*/*w*) was homogenized at room temperature for 1 h under constant magnetic stirring. The biopolymer solution was poured into an elliptically-shaped mold (22 mm length and 11 mm diameter) and dried in a Memmert UM 400 drying oven (Schwabach, Germany) at 25 °C with a relative humidity of 40% for 1 h. Subsequently, 2 mL of sacha inchi oil was injected into the formed soft capsule. The syringe hole in the capsule was then sealed by carefully applying heat using a small hot spatula.

### 2.5. Fatty Acids’ Polyunsaturated Bioaccessibility

The encapsulated oil was submitted to an in vitro digestion process using mouth, gastric, and intestinal simulation. This method was implemented as described by Pacheco et al. [31]. To simulate mouth digestion, 5 g of soft capsules were weighed and added to 9 mL of simulated saliva (1.59 mM CaCl_2_, 21.1 mM KCl, 14.4 mM NaHCO_3_, 0.2 mM MgCl_2_, pH adjusted to 7.0 using 1.0 N HCl, and the α-amylase enzyme). This mixture was incubated in a Schutzart DIN 60529-IP 20 shaking water bath (Memmert, Germany) for 5 min at 37 °C, and agitated at 185 rpm. Gastric digestion was then initiated by adding 36 mL of pepsin solution (25 mg/mL in 0.02 N HCl) to the samples. The mixture’s pH was adjusted to 2.0 using 1.0 N HCl, and was incubated while continuously shaking at 130 rpm at 37 °C for 60 min. To simulate intestinal digestion, the gastric-digested mixture’s pH was adjusted to 6.0 using 1 M NaHCO_3_. Afterwards, 0.25 mL of pancreatin solution (2 g/L) and biliary salts (12 g/L) dissolved in aqueous 0.1 M NaHCO_3_ were added and then the mixture was incubated at 37 °C for 120 min while constantly stirring at 45 rpm.

After intestinal digestion, the samples were immediately centrifuged at 5000 rpm for 10 min. The centrifuged samples were separated into two layers: an opaque sediment layer at the bottom, and a thin oily layer at the top. The oily layer was centrifuged again at 4000 rpm for 3 min and filtered using a Millipore membrane (0.45 μm), and then analyzed for its polyunsaturated fatty acid content and composition using GC–FID and GC–MS. The percentage of bioaccessibility was calculated using this equation:(1)$$\mathrm{Bioaccessibility}(\%)=\frac{\left(\mathrm{Content}\;\mathrm{of}\;\mathrm{fatty}\;\mathrm{acids}\;\mathrm{polyunsaturated}\;\mathrm{present}\;\mathrm{in}\;\mathrm{the}\;\mathrm{digestion}\;\mathrm{product}\right)}{\left(\mathrm{Content}\;\mathrm{of}\;\mathrm{fatty}\;\mathrm{acids}\;\mathrm{polyunsaturated}\;\mathrm{present}\;\mathrm{in}\;\mathrm{the}\;\mathrm{encapsulted}\;\mathrm{matrix}\right)}\times 100$$

## 3. Results

### 3.1. Characterization of Sacha Inchi Oil

#### 3.1.1. Fatty Acid Identification with Gas Chromatography Coupled to Mass Spectrometry (GC–MS)

As shown in the GC–MS chromatograms in Figure 1, the fatty acid profiles revealed 13 main chemical constituents in the SIO sample prior to encapsulation. The peaks numbered 1, 2, and 3 in the oil, eluted at retention times of 31.34, 33.15, and 33.19 min, were identified as hexadecanoic acid methyl ester, 9,12–octadecadienoic acid methyl ester, and (Z,Z,Z)-9,12,15-octadecatrienoic acid methyl ester, respectively. This demonstrates that the oil is a rich source of polyunsaturated fatty acid. Peak number 4 in the chromatograms, eluted at a retention time of 33.52 min, was identified as (Z)-11-octadecenoic acid methyl ester and octadecenoic acid methyl ester. The peaks numbered 5, 6, 7, 8, and 9 were observed in the oil eluted at times of 33.52, 36.91, 37.68, 38.41, and 39.11 min, respectively, and were identified as eicosane. In the sample, the compounds hexadecane (peak number 10), 1,4-phthalazinedione 2,3 dihydro-6-nitro (peak number 11), cyclotrisiloxane hexamethyl (peak number 12), and 5-methyl-2-phenylindolizine (peak number 13) were eluted at retention times of 39.79, 40.45, 41.08, and 41.74 min, respectively.

#### 3.1.2. Fatty Acid Profile with Gas Chromatography with a Flame Ionization Detector (GC–FID)

The fatty acid contents analyzed by GC–FID in the unencapsuslated SIO sample (i.e., prior to encapsulation) are shown in Table 1. The presence of α-linolenic (C18:3 ω-3), linoleic (C18:2 ω-6), palmitic (C16:0), stearic (C18:0), and oleic (C18:1 ω-9) acids, in decreasing order of abundance, indicated a rich source of polyunsaturated fatty acids. These values agreed with the results of Chirinos et al. [3] and Gutiérrez et al. [32]. The content of ω-3 (59.23 g/100 g) is important and desirable because of its contribution to preventing several diseases such as obesity, diabetes, allergies, Alzheimer’s, and coronary and neurodegenerative diseases [33]. The ω-6/ω-3 ratio of 0.56 has many health and nutritional benefits, such as the reduction of chronic diseases, and in cardiovascular and hypertension disease prevention [34,35]. The low ratio of linoleic/α-linolenic is similar to the results obtained in both irradiated and non-irradiated sacha inchi oils [32].

### 3.2. Chemical Content of Encapsulated SIO after Simulated In-Vitro Digestion

In order to study the impact of soft capsules on the bioaccessibility of the fatty acids, the encapsulated sacha inchi oil was subjected to in vitro digestion (i.e., mouth–gastric–intestinal media simulation), and its chemical content was analyzed using GC–MS and GC–FID spectrometry.

#### 3.2.1. Products Derived from the Degradation of Polyunsaturated Fatty Acids after In Vitro Digestion

The chromatogram profile of the main SIO compounds and their degradation by-products using GC–MS showed a maximum of 17 chemical constituents, as displayed in Figure 2. No peak attributable to 9,12–octadecadienoic acid methyl ester was detected in the chromatograms of the encapsulated oil after digestion, and therefore its bioaccessibility was considered negligible. Conversely, the presence of (Z,Z,Z)-9,12,15-octadecatrienoic acid methyl ester was detected before and after in vitro digestion in peaks number 3 and 13, which were eluted at a retention time of 33.20 and 32.06 min, respectively. The presence of this fatty acid was detected in the chromatograms of the encapsulated oil after digestion in a 43.96% peak area, making its bioaccessibility considerably significant. The presence of peaks with different retention times was also evidenced. For instance, 10,13-octadecadienoic acid methyl ester was identified as a degradation by-product of 9,12–octadecadienoic acid methyl ester [36], along with derivatives such as propanal, alcoholic amines, and aromatic compounds. All were related as by-products of simulated gastrointestinal digestion, demonstrating a low bioaccessibility of the encapsulated oil.

#### 3.2.2. Fatty Acid Content under the Simulated In Vitro Digestion

The relative concentration of the encapsulated fatty acids before and after the simulated gastrointestinal digestion, determined using GC–FID spectrometry, is shown in Figure 3. It was found that the bioaccessibility of the α-linolenic, linoleic, and oleic polyunsaturated fatty acids was 1.70%, 1.46%, and 35.8% respectively, while the saturated stearic and palmitic acids presented bioaccessibility values of 2.26% and 1.72%, respectively.

## 4. Discussion

As seen in Figure 3, during the digestion of the SIO encapsulated in G/CM softgels, there was a significant decrease in the content of the two most abundant PUFAs (i.e., α-linolenic and linoleic acid content), which entailed a reduction in the nutritional and functional value of this natural oil. As can be inferred by comparing the GC–MS chromatograms of the SIO before and after in vitro digestion (Figure 2 and Figure 3), the amounts of unsaturated fatty acids originally present in the SIO stimulated a higher generation of oxidation by-products during digestion [28]. Similar results were reported by Nieva-Echevarría, Goicoechea, and Guillén [37] during the in vitro gastrointestinal digestion of flaxseed oil. The amount of non-encapsulated oil (i.e., the oil that remained on the surface of the capsule) may also have affected the stability of the bioactive compound. In other words, the oil that was present at the surface of the capsule might have undergone oxidation and thereby affected the oxidative stability of the sample [38]. Alpizar-Reyes et al. [39] observed a similar situation when sesame seed oil was microencapsulated with tamarind seed mucilage.

Compared with the free sacha inchi oil [40], during in vitro digestion, the soft capsules showed a low protection of the encapsulated PUFAs against gastric conditions because of the nature of the wall materials and the G/CM ratio in the matrix. This behavior was associated with the intermolecular interactions between the functional groups of CM combined with G, which gradually disappeared as a result of the repulsion forces of the biopolymers [21,41,42], as well as the rapid degradation of the gelatin by the proteolytic enzymes present in the stomach [20], which reduced the barrier properties of the matrix. This was insufficient at protecting the oil against oxidation, and in turn affected the porosity of the biopolymer matrix and the oil release rate during in vitro digestion. Furthermore, the low amount of CM hydrocolloid in the soft capsule allowed for the degradation and facilitation of the enzymatic hydrolysis of G under gastric conditions [40].

These results seem to contradict the results of other published studies. Cortés-Camargo et al. [43] found that there was a delayed release of lemon essential oil microencapsulated using mesquite gum–chia mucilage mixtures. Papillo et al. [44] reported a high in vitro bioaccessibility of curcuminoids that were microencapsulated using gum arabic and maltodextrins as encapsulating agents. Da Silva Stefani et al. [45] reported a good bioavailability of nanoencapsulated linseed oil using chia seed mucilage as a structuring material. Jannasari et al. [42] studied the microencapsulation of vitamin D using gelatin and cress seed mucilage, and found release rates in the gastric and intestinal media of 28% and 70%, respectively. In addition, Barrow et al. [46] reported a high bioavailability of omega-3 fish oil microencapsulated using the technique of complex coacervation (140–180 mg EPA/DHA/g powder) in contrast with softgel capsules (data not shown). These results suggest that the G/CM matrix is not an encapsulating biopolymer that is sufficiently resistant to gastric conditions, thus reducing the bioaccessibility of the bioactive compounds carried by G/CM softgels.

## 5. Conclusions

In this work, oil samples extracted from sacha inchi seeds were encapsulated in softgels composed of gelatin (G) and cactus mucilage (CM) biopolymers, and then exposed to simulated gastric conditions. The nutritional composition of the oil samples was evaluated before and after in vitro digestion by means of GC–MS and GC–FID spectrometry. In this way, the protective capacity of the contents of sacha inchi oil offered by the G/CM biopolymeric wall of the softgel against digestive processes was evaluated.

α-Linolenic (C18:3 ω-3), linoleic (C18:2 ω-6), oleic (C18:1 ω-9), stearic (C18:0), and palmitic (C16:0) acids were the main fatty acids present in the non-encapsulated sacha inchi oil. It was found that the content of polyunsaturated fatty acids (PUFAs), especially α-linolenic (C18: 3 ω-3) and linoleic (C18: 2 ω-6), carried by the G/CM softgels, decreased significantly during in vitro digestion (bioaccessibility equal to 1.70% or 1.46%, respectively), which revealed a reduction in the nutritional value of the encapsulated oil after undergoing gastric processes. The low protective capacity of the G/CM wall material was attributed to the low concentration of the CM hydrocolloid, which left the gelatin biopolymer exposed to enzymatic hydrolysis.

Although the bioaccessibility of the PUFAs obtained was relatively low, we believe that the use of a mixture of proteins (gelatin) and heteropolysaccharides (cactus mucilage) for the manufacture of microcapsules can act as a suitable delivery system for the incorporation of other bioactive compounds within acidic food matrices before being subjected to digestive processes, for example, by encapsulating functional agents and subsequent release (by shacking) in media such as fruit juices or dairy drinks. This result will undoubtedly be interesting for certain applications in the food and pharmaceutical industries.

## Figures and Tables

**Figure 1 polymers-12-01995-f001:**
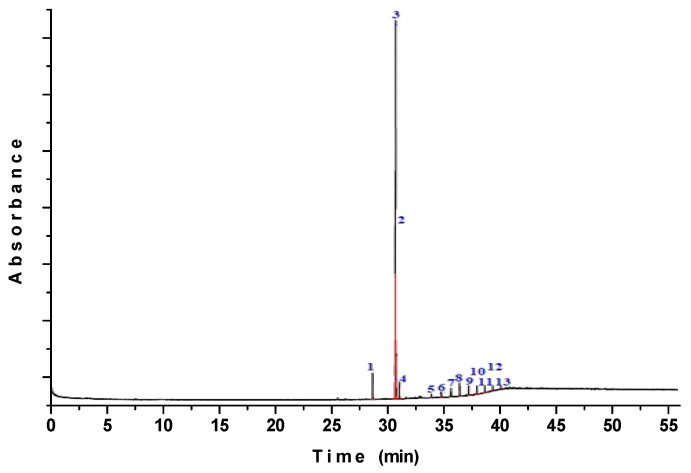
GC–MS chromatograms of sacha inchi oil before the simulated gastrointestinal digestion.

**Figure 2 polymers-12-01995-f002:**
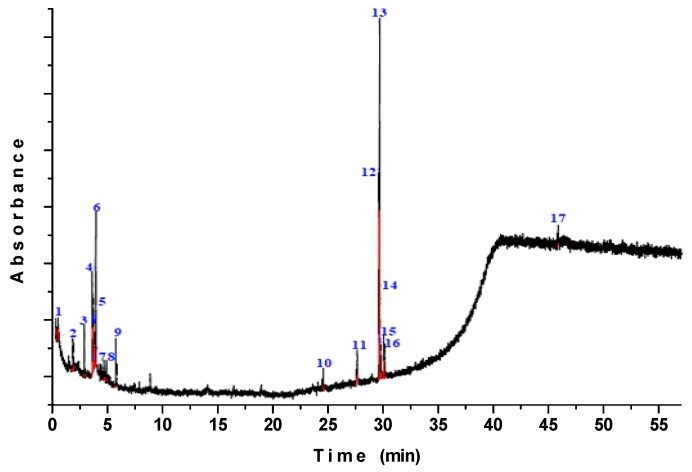
GC–MS chromatograms of sacha inchi oil after the simulated gastrointestinal digestion. Abbreviations: (**1**) Ethanol 2-(2-methoxyethoxy); (**2**) 3-[2-Diethylaminoethyl]-2,4-pentanedione; (**3**) Diethyl carbamoyl *t*-butoxy sulfide; (**4**) Ethane 1,2-bis(methylthio); (**5**) Ethane 1,2-bis(methylthio); (**6**) Cyclotetrasiloxane octamethyl; (**7**) Cycloserine; (**8**) 1-Propanol 2-amino; (**9**) 2-Pentanamine *N*-(1-methylbutyl); (**10**) *n*-Hexylmethylamine; (**11**) Methylpent-4-enylamine; (**12**) 10,13-Octadecadienoic acid methyl ester; (**13**) (Z,Z,Z)-9,12,15-Octadecatrienoic acid, methyl ester; (**14**) Epinephrine; (**15**) 1-Octanamine *N*-methyl; (**16**) 2-Amino-1-(*o*-hydroxyphenyl)propane; (**17**) Tetrasiloxane, decamethyl.

**Figure 3 polymers-12-01995-f003:**
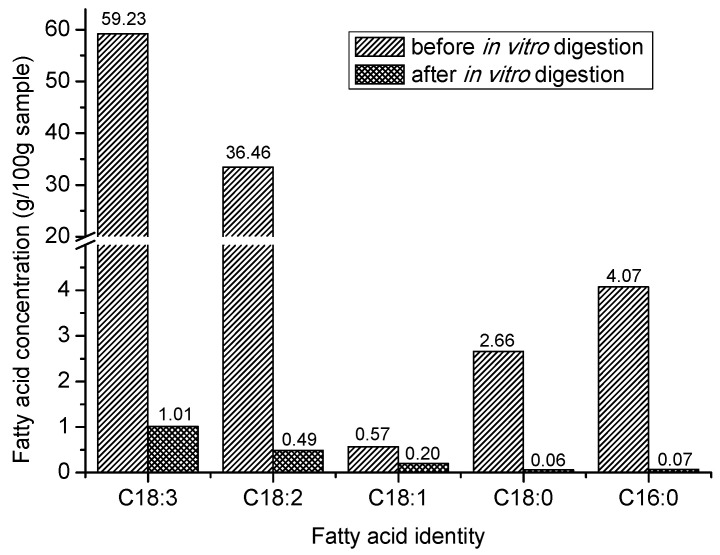
Relative concentration of fatty acids in sacha inchi oil (SIO) samples before and after the simulated gastrointestinal digestion.

**Table 1 polymers-12-01995-t001:** Fatty acid composition (g/100 g) of sacha inchi oil before the simulated gastrointestinal digestion.

Fatty Acid	g/100 g-Sample
α-Linolenic (C18:3 ω-3)	59.23
Linoleic (C18:2 ω-6)	33.46
Oleic (C18:1 ω-9)	0.57
Stearic (C18:0)	2.66
Palmitic (C16:0)	4.07

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
