# Peer review of "Encapsulation Effect on the In Vitro Bioaccessibility of Sacha Inchi Oil (Plukenetia volubilis L.) by Soft Capsules Composed of Gelatin and Cactus Mucilage Biopolymers"

_polymers, 2020, doi:10.3390/polym12091995_

Round 1

Reviewer 1 Report

Plukenetia volubilis L. seed oil is rich in polyunsaturated fatty acids, whose nutritional efficacy is limited due to their low water solubility and labile bioaccessability. In order to address this issue, Plukenetia volubilis L. seed oil is encapsulated in the soft capsules composed of gelatin and cactus mucilage biopolymers. Afterwards, the bioaccessability of Plukenetia volubilis L. seed oil subjected to the in-vitro simulated digestive process is evaluated. This study points out that the encapsulation can limit the in-vitro bioaccesibility of Plukenetia volubilis L. seed oil, which is valuable for the application of Plukenetia volubilis L. seed oil in the food and pharmaceutical industry.

The experiments of this study are well designed, and the writing is well organized. The existing issues and the approaches of addressing them are clearly and concisely described in the Introduction section, and the experimental results are fully discussed. The positive results are obtained. The subject of this study is applicable to Polymers. I think that this manuscript can be accepted for publishing. The following issues are suggested to be addressed as authors revise this manuscript.

Lines 27-28: “Therefore, the soft capsules are able to limit the in vitro bioaccesibility of PUFAs due the nature of the wall materials and G/CM ratio in the matrix”: due? due to ?

Lines 36-38: “The percentage of edible oil in sacha inchi seeds ranges from 41 to 54% (w/w) and contains a high proportion of lipids (35-60%), free fatty acids (1.2%), and phospholipids (0.8%) [2].”: I suggest authors to improve these descriptions in terms of grammar.

Line 50: “and its ability to easy dissolve in biological media”: I suggest authors to improve this description (to easy dissolve).

Lines 130-142: 2.4. Soft capsule preparation with SIO inclusion: After reading this section, I can not understand which step sacha inchi oil (SIO) is added. Additionally, when the encapsulation is formed, the pH of solution is adjusted?

Reviewer 2 Report

The authors reported the preparation of a softgel capsules based on gelatin and cactus-mucilage ingredients for in vitro digestion of encapsulated sacha inchi oil (SIO). Two gas chromatography techniques were applied (GC-MS and GC-FID) to evaluate the bioaccesibility of SIO by quantify and identify bioactive compounds before and after simulated digestion process.

Abstract

Table 1 shows fatty acid composition in units of g/100g for both before and after in vitro digestion. However, the abstract shows different values after digestion, Additionally, these values are presented in %.

Introduction

It is necessary to describe the concept of bioaccesibility in order to differentiate from bioaccesibility.

L80.- Describe what GC-MS and GC-FID stands for

Materials and methods

L103.- Delete “The quantification and identification of fatty acids presented in SIO are shown in section 2.3.

L141.- Explain how the 2mL of SIO were incorporated into the soft capsule?

L147.- Determination of polyunsaturated fatty acids in samples is not clear:

How the sample was obtained for this measurement, was it from softgels with oils or just the softgels?

How the sample was obtained before the in vitro digestion?

Results

L261.- Figure 3 is expressed in g/100g, however, the discussion states values after digestion in (%). Also, these values do not correspond with table 1.

All figures and tables are poorly recalled. They need more discussion oriented to the polymer science.

Discussions

L273.- Non-encapsulated oil?. This treatment was not included in material and methods section.

L291-293. Rewrite the sentence.

Conclusions

L301-309.- Rewrite the conclusions. Experimental work for determination of fast release of encapsulated active ingredients was not carried out in this study.

Round 2

Reviewer 2 Report

No comments